# Influence of the ZrO₂ Crystalline Phases on the Nature of Active Sites in PdCu/ZrO₂ Catalysts for the Methanol Steam Reforming Reaction—An In Situ Spectroscopic Study

**Daniel Ruano** [1,2], **Beatriz M. Pabón** [2], **Càtia Azenha** [3], **Cecilia Mateos-Pedrero** [3],
**Adélio Mendes** [3], **Virginia Pérez-Dieste** [1,*] **and Patricia Concepción** [2,*]

[1]  ALBA Synchrotron Light Source, Carrer Llum 2-26, 08290 Cerdanyola Del Valles, Spain; druano@cells.es
[2]  Instituto de Tecnología Química, Universitat Politècnica de València-Consejo Superior de Investigaciones Científicas (UPV-CSIC), Avenida de los Naranjos s/n, 46022 Valencia, Spain; Bmpabon@ucm.es
[3]  LEPABE—Laboratory for Process Engineering, Environment, Biotechnology and Energy, Faculty of Engineering, University of Porto, Rua Dr. Roberto Frias, 4200-465 Porto, Portugal; catia@fe.up.pt (C.A.); cmpedrero@fe.up.pt (C.M.-P.); mendes@fe.up.pt (A.M.)
*  Correspondence: vperezdieste@cells.es (V.P.-D.); pconcepc@upvnet.upv.es (P.C.)

**Abstract:** In this work, the electronic properties of the metal sites in cubic and monoclinic ZrO₂ supported Pd and PdCu catalysts have been investigated using CO as probe molecule in in-situ IR studies, and the surface composition of the outermost layers has been studied by APXPS (Ambient Pressure X-ray Photoemission Spectroscopy). The reaction products were followed by mass spectrometry, making it possible to relate the chemical properties of the catalysts under reaction conditions with their selectivity. Combining these techniques, it has been shown that the structure of the support (monoclinic or cubic ZrO₂) affects the metal dispersion, mobility, and reorganization of metal sites under methanol steam reforming (MSR) conditions, influencing the oxidation state of surface metal species, with important consequences in the catalytic activity. Correlating the mass spectra of the reaction products with these spectroscopic studies, it was possible to conclude that electropositive metal species play an imperative role for high $CO_2$ and $H_2$ selectivity in the MSR reaction (less CO formation).

**Keywords:** APXPS; Near Ambient Pressure-X-ray Photoelectron Spectroscopy; X-ray spectroscopy; in situ characterization; copper; palladium; CuPd alloy; methanol steam reforming; zirconium oxide; infrared spectroscopy

## 1. Introduction

Catalysis plays an important role in the industrial chemistry. In spite of being a longstanding discipline, fundamental knowledge of reaction mechanisms and the active sites—essential for developing new catalysts with improved performance—is missing for numerous catalytic processes. In this framework, spectroscopic characterization of catalysts under reaction conditions is an important research field [1], where new and improved technologies with better temporal and spatial resolutions continuously emerging, enabling more accurate identification of active species and reaction intermediates [2]. The dynamic nature of catalysts, which tend to adapt to the reaction environment—experiencing remarkable morphological and compositional and modifications affecting their catalytic activity—cannot be ignored [3,4]. The combination of multiple spectroscopic tools for studying catalysts under 'in situ' (i.e., operando) or near to real conditions is essential

for a comprehensive picture of the active sites under steady state or transient conditions [5–7]. Moreover, simultaneous detection of reaction products by on-line gas chromatography (GC) or mass spectrometry (MS) in combination with the spectroscopic study is highly recommended in order to achieve appropriate structure–activity correlations. Besides its bulk properties, the surface composition and the nature of surface sites on the upper layers of a catalyst are key parameters to explain the catalytic process, hence the high importance of surface sensitive spectroscopic tools. X-ray photoelectron spectroscopy (XPS) is one of the most powerful techniques to investigate the chemical state of elements on the surface of solid materials [8]. The instrumental adaption to supress the requirement of high vacuum in the XPS analysis chamber made it possible to analyze the state of catalysts under gaseous sample environments which—while still far from industrial pressure conditions, due to the intrinsic pressure limitation of soft X-rays ambient pressure XPS (maximum 130 mbar, typically around 1–20 mbar) [9,10])—could extract valuable information about the state of the catalyst under different reaction environments. Another interesting and complementary spectroscopic tool is infrared (IR) spectroscopy, which gives information of the catalyst surface with the possibility to analyse the state and electronic properties of surface sites under different environments, i.e., from UHV up to high pressure (~20–30 bar) [11]. Moreover, working under reaction conditions, it is possible to identify kinetically relevant active sites and reaction intermediate species [12].

In this work, the nature of surface species and their dynamics in metal supported $ZrO_2$ catalysts under methanol stream reforming (MSR) conditions have been studied combining in situ XPS and IR spectroscopy. Simultaneous MS can correlate spectroscopic features with the catalytic performance of the studied catalysts. MSR is a promising technology for $H_2$ production for fuel cell applications. Even though $H_2$ can be produced using other fuels, like ethanol or methane, methanol is the simplest one, as the absence of strong C–C bonds favors low temperatures of steam reforming operation (200–250 °C), in contrast to that of the other compounds (400–500 °C) [13]. In addition, it is biodegradable, has a high hydrogen to carbon ratio, and is liquid at atmospheric conditions. One important aspect in the MSR process is that CO generation needs to remain below 10 ppm, to avoid poisoning of the fuel cell anode. In this respect, the selection of a catalyst working efficiently at low operating temperatures, below 200 °C—in order to suppress side reactions like methanol decomposition (MD)—is essential. In our recent study [14], we reported a highly active and selective (4 wt %) Pd (20 wt %) Cu/$ZrO_2$-m MSR catalyst. In that work, catalytic data supported by conventional physical–chemical catalyst characterization (XRD, TPR-$H_2$, XPS, SEM-EDX) show differences in metal dispersion among the two catalysts, which result in different activity and selectivity. In this work, we have used operando spectroscopies to confirm those findings and to gain more insight into the electronic state of the active sites.

## 2. Results

$ZrO_2$ has widely been used as support in many catalytic processes [15,16]. It has three polymorphs (monoclinic, cubic, and tetragonal) [15] presenting different textural properties (i.e., surface area, micro-porosity) and Lewis acid sites, influencing the catalytic activity of the catalyst. These ends, on one hand, in different nucleation sites influencing the metal dispersion and, on the other hand, in dissimilar metal support interaction reverberating in metal stability and/or dynamic evolution of supported metal species under working conditions. In the next part, the influence of the monoclinic and cubic $ZrO_2$ phase on the dispersion of palladium in monometallic and palladium and copper in bimetallic catalysts will be analyzed, with special attention on the stability and/or dynamic behavior of the metal sites under MSR conditions and on their electronic properties affecting the MSR selectivity.

The catalytic and morphologic characterization of the monometallic (4 wt %) Pd/$ZrO_2$ and bimetallic (4 wt %) Pd (20 wt %) Cu/$ZrO_2$ catalysts was presented in our previous work [14]. Some details can be found in the Supplementary Materials.

### 2.1. Post-Reaction UHV and In Situ XPS Surface Characterization of Pd-Based Catalysts

The catalysts were first characterized by XPS using a lab-based XPS setup with a Al anode X-ray source ($h\nu$ = 1486 eV). The XPS UHV system is equipped with an annex high pressure catalytic reactor (HPCR), which can perform catalytic reactions at pressures up to 20 bar. After the reaction, the reactor is pumped down and the sample is transferred in UHV to the analysis chamber.

In the lab-based experiments, the samples were first reduced in the high-pressure cell at 1 bar and 300 °C and they were transferred to the analysis chamber for post-reduction characterization (in UHV). In a second step, the MSR reaction was performed at 180 °C and 1 bar in the high-pressure cell and the samples were also characterized post-reaction.

In order to achieve higher surface sensitivity and with the aim to work under in-situ conditions, APXPS experiments were carried out at the Near Ambient Pressure Photoemission branch of CIRCE beamline in ALBA synchrotron.

Both the monometallic and bimetallic samples were characterized post-reaction in the lab setup, while the synchrotron APXPS experiments focused on the monoclinic and cubic bimetallic samples.

The UHV XPS post-reduction and post-reaction spectra for the $Pd/ZrO_2$ monometallic and $PdCu/ZrO_2$ bimetallic catalyst are shown in Figure S1 and Figure 1, respectively, and the binding energies (BE) and the molar ratios calculated from the XPS fittings included in Table S1 and Table 1, respectively.

**Table 1.** $Pd3d_{5/2}$, $Cu2p_{3/2}$ and $Zr3p_{3/2}$ binding energy (BE, eV) and surface chemical composition (atomic ratio) of monoclinic $PdCu/ZrO_2$ and cubic $PdCu/ZrO_2$ catalysts

| Sample | Proc. | $Pd3d_{5/2}$ $(Pd^0)$ | $Pd3d_{5/2}$ $(Pd^{n+})$ | $Pd^{n+}/Pd^0$ | $Cu2p_{3/2}$ Cu (I) | $Cu2p_{3/2}$ Cu (II) | $Zr3p_{3/2}$ | Pd/Zr | Cu/Zr | Pd/Cu |
|---|---|---|---|---|---|---|---|---|---|---|
| PdCu ($ZrO_2$-m) | $H_2$ | 335.4 | – | – | 932.5 | – | 333.3 | 0.04 | 0.54 | 0.03 |
| | MSR | 335.4 | – | – | 932.5 | – | 333.3 | 0.02 | 0.56 | 0.08 |
| PdCu ($ZrO_2$-c) | $H_2$ | 335.8 | 338.0 | 0.18 | 932.3 | 935.3 | 336.8 | 0.23 | 1.13 | 0.20 |
| | MSR | 335.8 | 337.7 | 0.18 | 932.3 | 933.9 | 335.8 | 0.24 | 1.07* | 0.22 |

In the monometallic samples, a higher dispersion of palladium is detected in the reduced $Pd/ZrO_2$-m sample compared to the $Pd/ZrO_2$-c, based on their lower Pd/Zr molar ratio (Table S1) and in agreement with XRD data. Indeed, the XRD pattern of the $Pd/ZrO_2$-m sample shows no diffraction lines ascribable to the Pd metal, suggesting lack of long-range order, while crystalline metallic Pd with average particle size of 14.8 nm is detected in the $Pd/ZrO_2$-c sample (Figure S2a). Regarding the oxidation state of the palladium species, $Pd^0$ is predominately present in both samples (Figure S1), with a $Pd3d_{5/2}$ BE of 335.7 eV. After exposure of the catalysts to MSR conditions in the HPCR reactor, no appreciable changes are observed in either sample, preserving the Pd a metallic state, except for a slight Pd surface enrichment in the $Pd/ZrO_2$-c sample (Table S1).

In the case of the bimetallic $PdCu/ZrO_2$ samples, the molar ratios—i.e., Pd/Zr and Cu/Zr ratios (Table 1)—are larger in the cubic than in the monoclinic $ZrO_2$ supported catalyst, analogously to the monometallic samples, indicating surface metal segregation in the cubic catalyst. The XRD of the bimetallic $PdCu/ZrO_2$ samples (Figure S2b) shows $Cu^0$ in both samples with a larger particle size in the $PdCu/ZrO_2$-c sample (~51 nm versus 19 nm in the $PdCu/ZrO_2$-m sample), accounting for a lower dispersion [14], while no diffraction lines ascribed to Pd could be detected in any sample.

Pd species [17,18] were detected in the UHV–XPS spectra in both bimetallic samples, with slightly different BE. Thus, in both post-reduced and post-reaction samples the Pd3d is located at BE = 335.4 eV for the monoclinic $ZrO_2$ supported catalyst, and at BE = 335.9 eV for the cubic $ZrO_2$ one. Regarding the copper species, $Cu^0$ (BE 932.5 eV) is observed in both samples with some oxidized copper species (BE 935.3 eV after reduction and 933.9 eV after reaction) [19] in the $PdCu/ZrO_2$-c sample. As shown in Figure 1 and Table 1, a slightly high $Zr3p_{3/2}$ BE (BE = 336.8 eV and 335.8 eV after reduction and after reaction conditions, respectively) is observed in the cubic compared to the monoclinic base sample (BE = 333.4 eV). Such a high binding energy is likely due to charging of the $ZrO_2$ support

cubic sample favored by its inhomogeneous composition. Charging effects were not observed in the core levels corresponding to the metallic species which, after calibration, were consistently located within the expected BE. Therefore, shifts in the Zr core levels likely obey to differential charging of the insulating support.

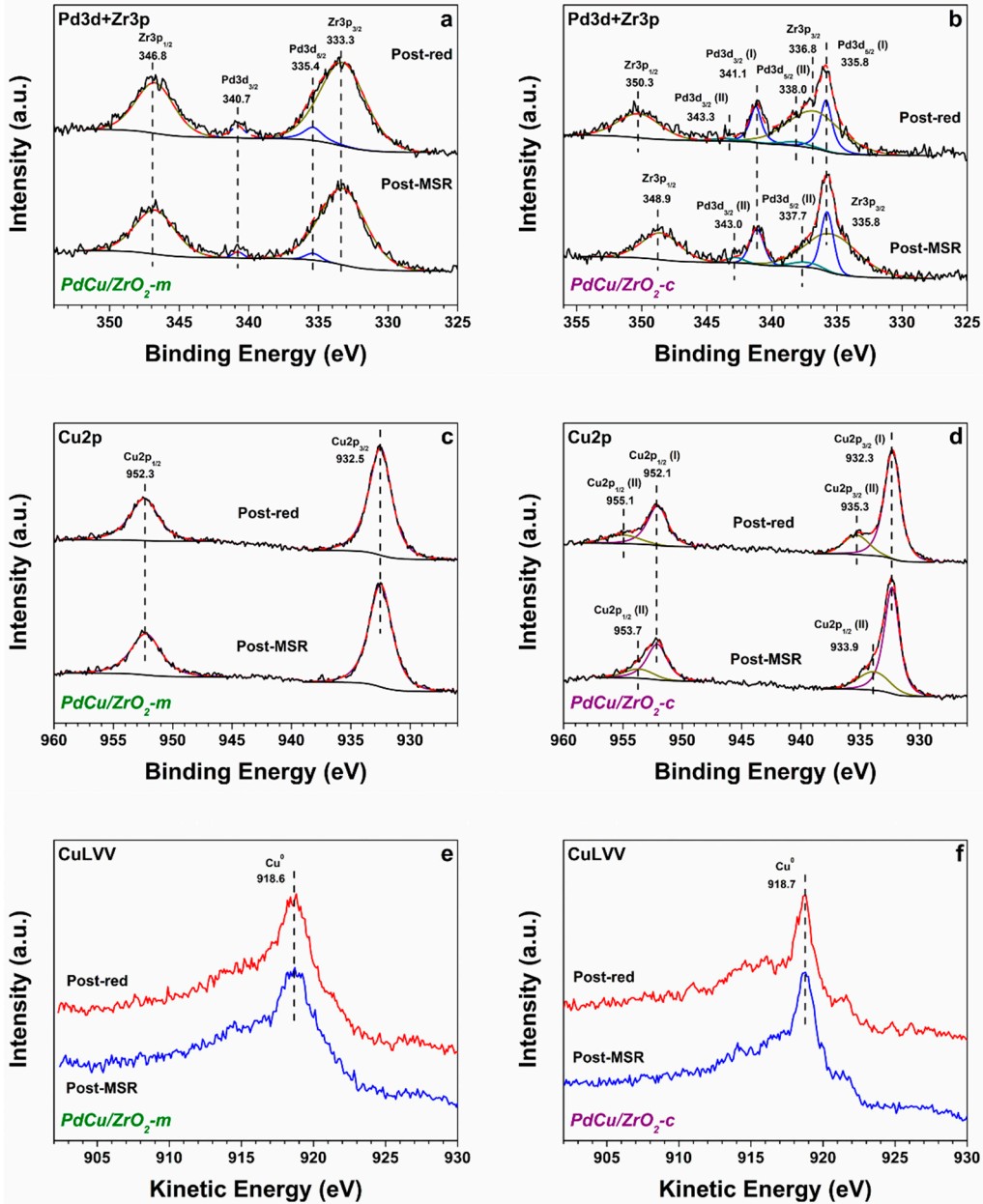

**Figure 1.** Pd3d + Zr3p core lines acquired with hν = 1486 eV of reduced (post-red) and after reaction (post-MSR) PdCu/ZrO$_2$-m catalyst (**a**) and PdCu/ZrO$_2$-c catalyst (**b**); Cu2p core lines of reduced (post-red) and after reaction (post-MSR) PdCu/ZrO$_2$-m catalyst (**c**) and PdCu/ZrO$_2$-c catalyst (**d**); CuLVV AES spectra of reduced (post-red) and after reaction (post-MSR) PdCu/ZrO$_2$-m catalyst (**e**) and PdCu/ZrO$_2$-c catalyst (**f**).

After methanol steam reformed in the high-pressure reaction cell, neither the binding energies, nor the chemical composition of either catalyst changed appreciably; however, noticeable structural and chemical modifications in the uppermost catalyst surface layers have been detected by synchrotron APXPS working at lower photon energy. The higher resolution and surface sensitivity at lower electron

kinetic energy and the in-situ conditions enable the identification of surface species which otherwise remain undetectable by post-reaction laboratory XPS.

Synchrotron APXPS experiments were performed with hν = 500 eV for Pd3d and Zr3p and hν = 1150 eV for Cu2p, corresponding to sampling depth around 1.6 nm, which is 2–4 times shorter than for the photon energy available in the laboratory XPS setup (3.7–6.4 nm).

The Pd3d and Zr3p core levels collected under reducing and MSR conditions are displayed in Figure 2a (PdCu/ZrO$_2$-m) and 2b (PdCu/ZrO$_2$-c), and the chemical composition of the PdCu/ZrO$_2$-m and PdCu/ZrO$_2$-c surfaces given in Table 2. The catalysts were first investigated under reducing conditions, in 1 mbar H$_2$ at 300 °C for 2 h. Then they were subsequently analyzed under MSR conditions, at a total pressure of 2.5 mbar and a molar ratio MeOH:H$_2$O = 1:1.5 (more information in the Materials and Methods section). Simultaneously to the XPS data, reaction products were analyzed by mass spectroscopy and the corresponding profiles displayed in Figure S3. The presence of reaction products was confirmed by O1s and C1s gas phase APXPS (Figure S4). A higher H$_2$/MeOH ratio is observed in the PdCu/ZrO$_2$-m sample versus PdCu/ZrO$_2$-c (see Table S3) and a higher CO/CO$_2$ ratio is observed in the PdCu/ZrO$_2$-c, in good agreement with the catalytic data reported in our previous work [14].

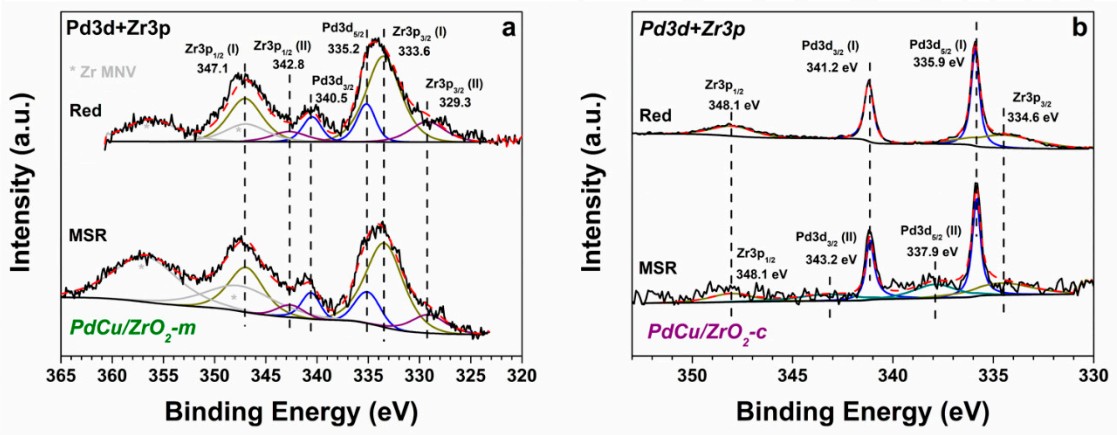

**Figure 2.** Pd3d + Zr3p core levels acquired with hν = 500 eV of reduced (**a**) and PdCu/ZrO$_2$-m and (**b**) PdCu/ZrO$_2$-c catalysts and under reaction conditions (MSR).

**Table 2.** Surface chemical composition (atomic ratio) of PdCu/ZrO$_2$-m and PdCu/ZrO$_2$-c catalysts during APXPS experiment.

| Sample | Proc. | Pd/Zr | Cu/Zr | Pd/Cu |
|---|---|---|---|---|
| PdCu ZrO$_2$-m | H$_2$ | 0.02 | 2.12 | 0.01 |
| | MSR | 0.02 | 1.11 | 0.02 |
| PdCu ZrO$_2$-c | H$_2$ | 0.28 | 3.52 | 0.07 |
| | MSR | 0.45 | 17.59 | 0.02 |

As can be seen in Figure 2, during reduction only metallic Pd (BE = 335.9 eV) is found in the cubic PdCu/ZrO$_2$. The Zr3p$_{3/2}$ signal is very weak and it appears centred at BE = 334.6 eV, which is around 1 eV higher than the values reported in literature for Zr$^{4+}$ [17,18]. During MSR, Pd partially oxidizes (BE 337.9 eV) and both Cu and Pd segregate towards the surface. The Pd/Zr atomic ratio increases from 0.28 to 0.45 and the Cu/Zr ratio increases from 3.52 to 17.59. As the reaction progressed, a substantial amount of hydrocarbon deposited on the surface, as indicated by the increase of the C1s component at 284.9 eV.

In the monoclinic ZrO$_2$ supported PdCu, the Pd3d$_{5/2}$ appears at BE = 335.2 eV during reduction (Figure 2 and Table 3). A species at that binding energy has been previously identified as PdCu alloy

with FCC structure in PdCu NPs aggregates [20]. The XRD pattern of the PdCu/ZrO$_2$-m sample (Figure S2b) shows a small non-well-defined component at 42.5° which could ambiguously be assigned to alloy species [14]. This component is not observed in the XRD pattern of the PdCu/ZrO$_2$-c sample, only Cu$^0$ was detected (Figure S2). While the binding energies for Pd3d5/2 in the lab data do not exactly coincide with APXPS ones, it should be noticed that also in the lab XPS the Pd3d$_{5/2}$ core level is located at lower values of binding energy in the monoclinic than in the cubic supported catalyst (335.4 vs. 335.8 eV), supporting the assignment of those two species as PdCu in the monoclinic ZrO$_2$ catalyst and Pd$^0$ in the cubic.

**Table 3.** Pd3d$_{5/2}$, Cu2p$_{3/2}$ and Zr3p$_{3/2}$ binding energy (BE, eV) and auger parameter of monoclinic PdCu/ZrO$_2$ and cubic PdCu/ZrO$_2$ catalysts during APXPS experiments

| Sample | Proc. | Pd3d$_{5/2}$ (I) (Pd$^0$) | Pd3d$_{5/2}$ (I) (Pd$^{n+}$) | Pd$^{n+}$/Pd$^0$ | Cu2p$_{3/2}$ (I) | Cu2p$_{3/2}$ (II) | Zr3p$_{3/2}$ (I) | Zr3p$_{3/2}$ (II) | αCu |
|---|---|---|---|---|---|---|---|---|---|
| PdCu (ZrO$_2$-m) | H$_2$ | 335.2 | – | – | 932.5 | 931.1 | 333.6 | 329.3 | 1851.4 |
|  | MSR | 335.2 | – | – | 932.5 | 931.1 | 333.6 | 329.3 | 1851.4 |
| PdCu (ZrO$_2$-c) | H$_2$ | 335.9 | – | – | 932.4 | – | 334.6 | – | 1851.3 |
|  | MSR | 335.9 | 337.9 | 0.58 | 932.4 | 936.0 | 334.6 | – | 1851.3 |

The Pd/Zr atomic ratio is much smaller (0.02 vs 0.28 for reduction and 0.02 vs 0.45 for MSR) in the monoclinic ZrO$_2$ catalyst than in the cubic one. This is also in good agreement with the lab-XPS results and with our previous studies indicating higher metal dispersion in the monoclinic system.

In contrast with the cubic PdCu/ZrO$_2$, the monoclinic system presents two species for the Zr3p$_{3/2}$, at 328.9 eV and 333.9, which might be due to different Zr oxidation states, see also the corresponding Zr3d spectrum in Figure S5 of Supplementary Materials [21], or to differential charging of the support. Two Zr MNV Auger transitions at KE = 144 eV and 153.3 eV are partially overlapping with the Pd-Zr spectra for hν = 500 eV [22]. Due to the lower surface Zr ratio, the Auger was not detected for the cubic ZrO$_2$-based sample. During reaction, the Auger transition at KE = 144 eV—which corresponds to Zr oxide—gains intensity. For comparison, a spectra at hν = 700 eV during MSR, without the Auger contribution is presented in Figure S6.

Regarding the copper species, Figure 3b shows the Cu2p core level during reduction and reaction, for the cubic ZrO$_2$ based catalyst. During reduction, the Cu is completely reduced, as confirmed by the Auger spectra (Figure 3d) and auger parameter (Table 3) [23]. During MSR, the Cu segregates to the surface, as indicated by the increase in the Cu/Zr atomic ratio (Table 2) from 3.52 to 17.59. Additionally, a new component appears at BE = 936.0 eV, which is likely due to hydroxylated Cu [20,24]. Coming back to the lab-based post-reduction and post-reaction XPS data, it can be seen in Figure 1d that a feature at 935.3 eV and 933.9 eV was also present in both the post-reduction and post-reaction spectra respectively, which could be due subsurface oxides persisting after an incomplete reduction. In order to confirm this assignment, depth profiling of Cu2p was performed in the APXPS studies under reduction conditions and the Cu subsurface oxide species at 935.5 eV was also detected, with hν = 1386 eV. It is worth noticing that the hydroxides component at 936.0 eV was only detected during the in-situ MSR reaction, at both probing depths, settling that that specie is formed under reaction conditions.

On the other hand, for the monoclinic PdCu/ZrO$_2$, it is remarkable that the Cu2p line presents an additional component to that at 932.5eV, located at 931.1 eV. This value is close to reported in literature for PdCu alloy [20], confirming the hypothesis deduced from the Pd3d binding energy and from the XRD data. In contrast with the cubic ZrO$_2$ supported catalyst, the Cu/Zr atomic ratio decreases in reaction conditions for the monoclinic one and no oxides or hydroxylated Cu species were detected.

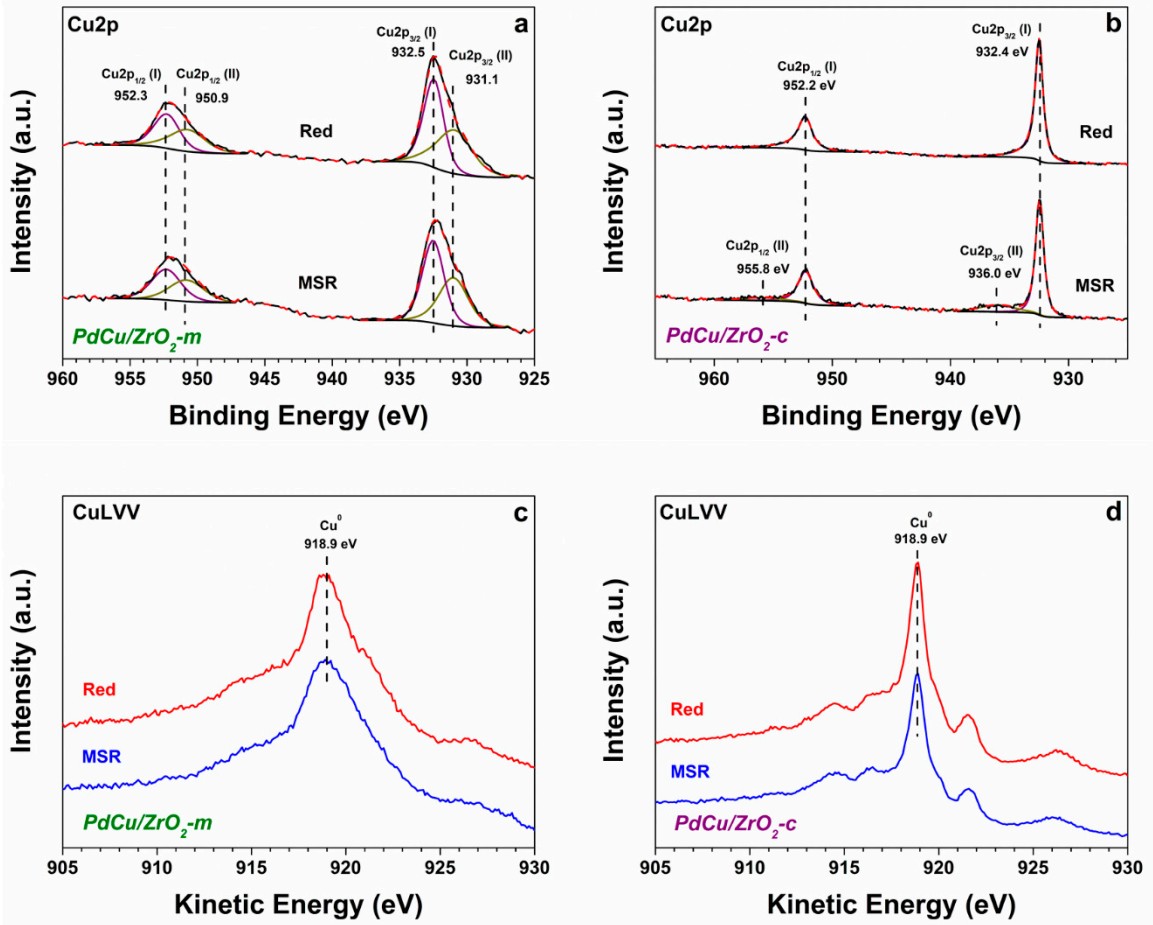

**Figure 3.** Cu2p core lines acquired with hν = 1150 eV of reduced (**a**) and PdCu/ZrO$_2$-m and (**b**) PdCu/ZrO$_2$-c catalysts and under reaction conditions (MSR). Auger AES spectra acquired with hν = 1150 eV under reduction and reaction conditions (MSR) of (**c**) PdCu/ZrO$_2$-m and (**d**) PdCu/ZrO$_2$-c catalysts.

In a general picture, it can be concluded that the cubic sample presented agglomerations of metal, both Pd and Cu, without evidence of alloying. Moreover, surface oxidation and/or hydroxylation is observed in both Pd and Cu metal phases respectively, accomplished by an important surface metal segregation under working conditions. In the cubic sample, the Fermi edge is clearly visible at the valence band spectrum (not included) and has the characteristic shape and binding energy of Cu3d. In contrast, PdCu alloy is present in the monoclinic catalyst and confirmed in both the Cu2p and Pd3d core levels, preventing in that case metal oxidation under reaction conditions. In addition, metal re-structuring is less pronounced in the monoclinic based catalyst.

## 2.2. Chemical State of the Pd-Based Catalysts Based on IR

Based on the above exposed data, metallic Pd and Cu species were observed in the reduced and working catalysts, which has been assigned as active sites in the MSR reaction. In this respect, a slight oxidation of surface Cu and Pd metal species is observed in the cubic ZrO$_2$ supported catalysts, accounting for their lower reactivity. However, an important aspect such as the electron charge density of surface metallic species and their exposed crystal facets, both of them with strong influence on the MSR selectivity, remain undetermined. IR spectroscopy using CO as probe molecule can provide information on these aspects, and most importantly, it can be done quasi 'in situ'. In this sense, the samples were reduced in situ in the catalytic IR cell and exposed to MSR conditions. After each process, the gas flow was stopped and the sample cooled down for CO titration (more details in

materials and methods section). CO is a very sensitive probe molecule to the chemical state of surface sites, both for Lewis acid sites and for metal sites [25–27].

In the metal free supports, after CO titration, IR bands at 2192 cm$^{-1}$, which is red shifted to 2184 cm$^{-1}$ at increasing CO coverage, 2173 and 2150 cm$^{-1}$ are formed in the $ZrO_2$-m (Figure S7a) and at 2165 and 2144 cm$^{-1}$ in the $ZrO_2$-c (Figure S7b). Those bands correspond to $Zr^{4+}$-CO complexes of $Zr^{4+}$ sites of different acid strength [25]. Based on the position of those bands (where IR bands at higher frequency corresponds to more acidic sites) and their relative intensity, a higher density and more diversity of Lewis acid sites in $ZrO_2$-m than in $ZrO_2$-c is inferred.

In the monometallic Pd reduced catalysts (Figure 4 and Table 4), the IR bands at 2192, 2172, 2165, and 2153 cm$^{-1}$, ascribed to the support are attenuated, indicative of their role as nucleation sites for metal interaction. Regarding the nature of palladium species, $Pd^+$ species are observed in both reduced samples characterized by IR at ~2135–2127 cm$^{-1}$. It is noteworthy that these species were not detected in the UHV XPS studies, probably due to their low amount. In addition, IR bands associated to linearly bonded CO on Pd (111) corners (2107–2089 cm$^{-1}$) and on edge sites of (111) and (100) facets (2065–2050 cm$^{-1}$) are observed, together with bridge-bonded CO on (111) and (100) planes (1982–1950 cm$^{-1}$) [28–30] and CO in a three-fold configuration (1877 cm$^{-1}$) [31,32]. Being the $\nu$(CO) frequency dependent on the electronic properties of surface sites, and based on the deconvolution of the IR spectra (Figure S8) different CO adsorption sites are observed in both Pd/$ZrO_2$-m and Pd/$ZrO_2$-c samples impairing dissimilar electron properties to the surface metal atoms. Again, XPS account for the same Pd BE on both samples, underlying the higher sensitivity of IR-CO for surface analysis of the electronic and local state of metal sites.

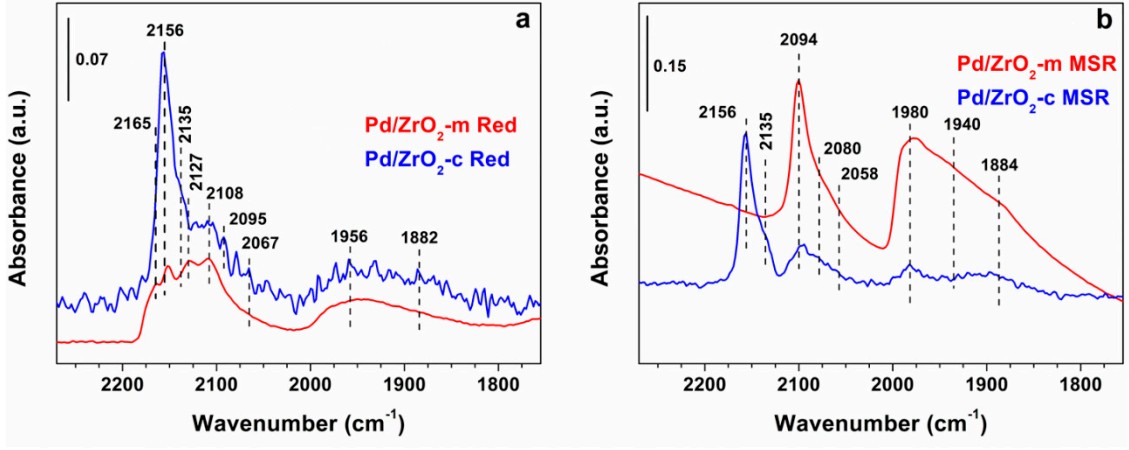

**Figure 4.** IR-CO adsorption on (**a**) reduced Pd/$ZrO_2$-m (red) and Pd/$ZrO_2$-c (blue); (**b**) after MSR. Included the position of the Pd-CO IR peaks obtained from spectra deconvolution.

**Table 4.** Position of the Pd-CO IR peaks obtained from spectra deconvolution of the catalysts Pd/$ZrO_2$-m and Pd/$ZrO_2$-c after reduction and after MSR.

| | Pd/$ZrO_2$-m (red) | Pd/$ZrO_2$-c (red) | Pd/$ZrO_2$-m (MSR) | Pd/$ZrO_2$-c (MSR) |
|---|---|---|---|---|
| CO on support ($Zr^{4+}$-CO) | 2165<br>2152 | -<br>2156 | -<br>- | -<br>2156 |
| $Pd^+$-CO | 2127 | 2118<br>2135 | - | 2135 |
| Linear CO on Pd (111) corners | 2108<br>2089 | 2095<br>- | 2094<br>2080 | 2100<br>2080 |
| Linear CO on edge sites of (111) and (100) | 2051 | 2067 | 2058 | 2056 |
| Bridge bonded CO on (111) and (100) | -<br>1956 | -<br>1951 | 1980<br>1940 | 1980<br>1912 |
| CO in three-fold configuration | 1892 | 1882 | 1884<br>1812 | 1863 |

However, when exposed to MSR conditions, both samples display important morphological changes due to mobility and surface reorganization of Pd species (Figure S9 and Table 4). Thus, in the Pd/ZrO$_2$-m sample, in addition to the reduction of Pd$^+$ species (disappearance of the band at 2127 cm$^{-1}$), the restructuration of the Pd species is reflected by a shift of the IR band at 2108 cm$^{-1}$ to 2094 cm$^{-1}$ and an increase in the intensity of the IR bands at 1980–1812 cm$^{-1}$, associated to bridge bonded CO and CO in a three-fold configuration of extended metal facets (Figure S9a). Sites due to the support (IR bands at 2192, 2172, 2165, and 2153 cm$^{-1}$) observed in the reduced sample, are not more visualized, being blocked by formate species with IR bands at 1573, 1388, and 1359 cm$^{-1}$ [33], generated under reaction conditions (Figure S10). Similar dynamic behavior is observed in the Pd/ZrO$_2$-c sample (Figure S9b), resulting in more defined IR bands at 2094, 2058, 1980, and 1882 cm$^{-1}$. In this case, formate species are not detected on the catalyst (Figure S10) surface explaining in that way the preservation of the IR band of the support (2153 cm$^{-1}$). Moreover, Pd$^+$ are not reduced under reaction conditions, due to the lower reducibility ascribed to the bigger Pd particle size. The most important feature is that, after being exposed to reaction conditions, both samples—in spite of presenting different initial surface topology—end up in similar exposed palladium facets (characterized by IR bands at 2100, 2080, 2058, 1980, 1940–1912, 1884–1863 cm$^{-1}$). Since the reactivity in MSR is strongly related to the electron properties of the metal site, and this is determined by the crystal topology, it becomes evident that both samples behaves analogous under reaction conditions displaying similar CO selectivity as reported in our previous work [14], while the activity of the Pd/ZrO$_2$-c is lower than that of the Pd/ZrO$_2$-m sample due to the presence of surface oxidised Pd$^+$ species in the former one. In this context, it is shown by IR spectroscopy using CO as probe molecule, how metal species move and reorganize under reaction conditions, changing the crystal morphology, i.e., exposed crystal facets. This behavior, not determined by lab-XPS, explain the already observed catalytic performance in the MSR reaction.

Addition of copper to the Pd based catalysts result in clear changes in the electronic state of metal surface sites as shown in Figure 5. In the reduced PdCu/ZrO$_2$-m sample, the IR bands at 2153, 2138, and 2112 cm$^{-1}$ are ascribed to CO interacting with the support, Cu$^+$ [34,35] and/or Pd$^+$, and metal species, respectively. Similar IR bands are observed in the PdCu/ZrO$_2$-c sample while the IR band assigned to the metal appears at lower frequencies (2098 cm$^{-1}$). The shift to high frequencies of the 2112 cm$^{-1}$ band respect to those usually ascribed to the metal site (~2107–2098 cm$^{-1}$) is associated to a more electropositive metallic character [25], i.e., less electron donation from the metal to the π* of CO, which can be related to electronic modifications in the metal particle due to alloy formation. Alloy formation was tentatively suggested in our previous work based on the XRD pattern and confirmed in this work based on APXPS studies. In addition, the IR data herein can identify a net positive charge associated to the PdCu alloy, which is hard to identified from the APXPS data. The positive charge of the metal particle undoubtedly influences the MSR selectivity.

Under MSR conditions, no significant changes are observed in the IR spectra of the bimetallic catalysts, which does not match with the NAP-XPS studies, where surface copper segregation was observed in the PdCu/ZrO$_2$-c sample. Probably the high light scattering (due to the black nature of the sample) and the low IR resolution, could explain this difference. On the other hand, on-line IR-MS analysis (Figure S11 and Table S4) agrees well with the activity of the catalysts in the catalytic studies, where a higher H$_2$ and CO$_2$ formation is detected in the PdCu/ZrO$_2$-m sample.

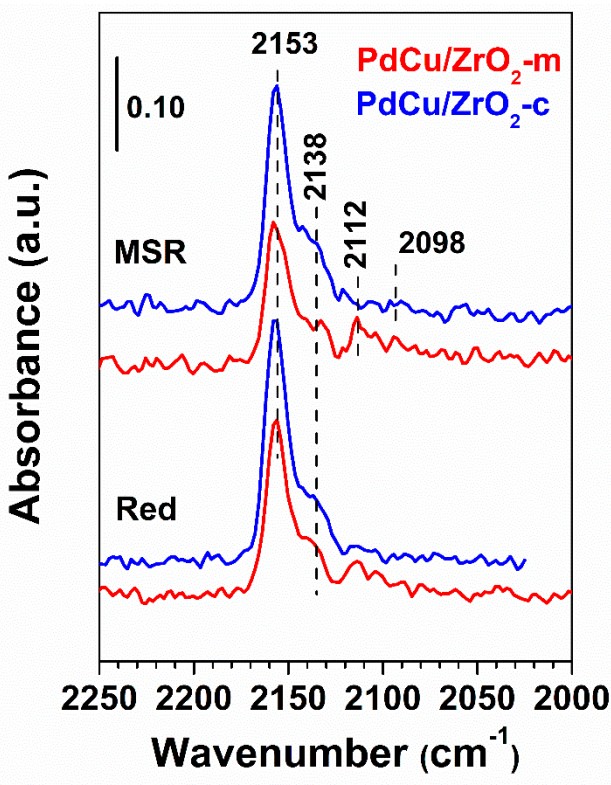

**Figure 5.** IR-CO adsorption on reduced (red) and after MSR PdCu/ZrO$_2$-c (blue) and PdCu/ZrO$_2$-m (red) samples.

## 3. Discussion

The herein XPS, APXPS, and IR-CO results show noticeable structural and chemical modifications in the uppermost catalyst surface layers in both monometallic (Pd) and bimetallic (PdCu) ZrO$_2$ based catalysts under MSR conditions. In fact, it has been detected that the palladium species in both Pd/ZrO$_2$-m and Pd/ZrO$_2$-c samples reorganizes under reaction conditions resulting in structures resembling similar exposed Pd metal faces. This fact explains previously reported results, indicating that both samples have similar catalytic performance in the MSR reaction. However, the presence of oxidised Pd species in the Pd/ZrO$_2$-c sample may account for their lower activity. This species hardly being detected by XPS suggests that their surface concentration is low. The addition of copper is shown to have an important effect on the electronic nature of surface exposed metal sites and on their stabilization, being strongly dependent on the nature of the support. Thus, a strong interaction between Cu and Pd and the support, resulting in improved metal dispersion, is observed in the case of monoclinic ZrO$_2$. Being both elements in a metallic state, a positive electronic character of the metal surface species is determined by IR-CO studies of in situ MSR exposed samples. This behavior is as consequence of the effective intermixing between both metals resulting in the formation of alloyed nanoparticles. Indeed, the XRD pattern of the PdCu/ZrO$_2$-m sample shows a broad component at 42.5°, which was tentatively assigned in our previous work to alloyed species. The formation of alloyed PdCu species is confirmed in this work by APXPS through the detection of Cu-Pd species both Cu2p and Pd3d core levels. Mass spectra analysis of the reaction products in the in situ spectroscopic studies, shows reduced CO formation in the PdCu/ZrO$_2$-m sample, which is related to the electropositive metal density charge of the PdCu alloyed species, playing a pivotal role for high selectivity in the MSR reaction with less CO formation. Notoriously, on average, the Pd/Cu molar ratio in the upper layers of the PdCu/ZrO$_2$-m catalyst (0.01 post-reduction and 0.02 post-MSR) is lower than that expected based on their nominal composition (i.e., 0.12), revealing heterogeneous distribution of Pd, which is located preferentially in the inner shell of the PdCu nanoparticle. It is suggested that the higher

electronegativity of Pd (2.2 versus 1.9 for Cu) favors the presence of positive charged Cu species on the particle surface, as detected by IR-CO. Conversely, in the case of cubic $ZrO_2$, the interaction between both metals is much weaker, appearing as segregate metal phases. In fact, diffraction lines associated to Cu were detected in the XRD pattern of the reduced samples. Under reaction conditions, surface palladium and copper species agglomerate in the surface, resulting in a strong increase of the surface Pd/Zr and Cu/Zr molar ratios, from 0.28 to 0.45 and from 3.52 to 17.59 respectively. As a result of a non-effective metal interaction, the electronic properties of the metal remain unaltered, and in consequence the above reported positive effect on the MSR reaction is not observed. Indeed, CO is detected by online mass spectroscopy. In addition, surface Pd oxidation and Cu hydroxylation is observed under MSR conditions, resulting in a loss of active sites with a decrease in the catalytic activity. Interestingly, different Zr oxidation states were detected in the PdCu/$ZrO_2$-m sample based on APXPS studies, which correspond to surface defects. The presence of surface defects plays an important role in water activation. This may account, together with the presence of alloyed species, for the improved catalytic activity observed in this sample. Thus, combining spectroscopic data with catalytic data, it has been demonstrated that positive charged metal species as a result of an effective intermixing between Cu and Pd, promote high selectivity in the MSR reaction with less CO formation.

## 4. Materials and Methods

### 4.1. Samples Preparation

The samples were prepared according to [14] and their physico-chemical properties are included on it.

The $ZrO_2$ supports were obtained by precipitation of $ZrO(NO_3)_2$ (Wako Chemicals) with $NH_4OH$. The appropriated yttrium nitrate amount was added in order to stabilize the different crystalline phases of $ZrO_2$: cubic (10 mol%) and monoclinic (no addition), respectively. The precipitate was recovered by filtration, dried at 100 °C and calcinated in static air at 600 °C for 5 h.

Monometallic catalysts containing 4 wt % of Pd were prepared by wet impregnation of the previously prepared supports. A solution of a calculated amount of palladium acetate ($Pd(O_2CCH_3)_2$; Sigma–Aldrich (Merck KGaA, St. Louis, Mo., USA.) was prepared with chloroform. Then the appropriate amount of support was added to the solution. The impregnated samples were dried at 120 °C overnight and calcined at 400 °C for 1 h in an oxygen atmosphere (60 mL·min$^{-1}$), cooled in $N_2$ (60 mL·min$^{-1}$) to room temperature, and then reduced in a hydrogen flow (150 mL·min$^{-1}$) for 2 h at 400 °C.

Bimetallic PdCu-supported catalysts containing 20 wt % Cu and 4 wt % Pd were obtained by sequential impregnation. First, palladium was loaded onto the $ZrO_2$ support, dried and heated under $O_2$ at 400 °C for 2 h. Then copper was impregnated on the Pd/$ZrO_2$ sample using copper nitrate ($Cu(NO_3)_2$; Prolabo) as precursor. These catalysts were calcined at 360 °C in static air for 8.5 h and then reduced in $H_2$ for 2 h at 400 °C. For comparative purposes, a monometallic Cu (20 wt % Cu) catalyst supported on the previously obtained $ZrO_2$-m carrier was prepared. The same experimental procedure as that used for the impregnation of Cu in the bimetallic PdCu/$ZrO_2$ catalysts was followed.

### 4.2. Laboratory Scale XPS

XPS spectra were recorder with a PHOIBOS 150 MCD-9 multichannel analyzer (SPECS GmbH, Berlin, Germany) using a non-monochromatic AlK$\alpha$ (1486.6 eV) X-ray source. The spectra were recorded in UHV (1·10$^{-9}$ mbar) using an X-ray power of 50 mW and an energy pass of 30 eV. The catalysts were pelleted (10 mg) and mounted into SPECS stainless steel sample holder. Before XPS analysis, different treatments were carried out over the catalyst in a high-pressure reactor (HPCR) directly connected to the main chamber under UHV. At first, the sample was reduced in the reactor using $H_2$ (10 mL·min$^{-1}$ flow) at atmospheric pressure and at 300 °C for 3 h. After reduction, the sample was transferred to the analysis chamber for XPS analysis. Following this, the catalyst was exposed

to MSR conditions at 180 °C and at atmospheric pressure for 2 h using a molar ratio MeOH:$H_2O$ of 1:1.5. MeOH (17 mL·$min^{-1}$ MeOH/Ar) and water (17 mL·$min^{-1}$ $H_2O$/Ar) were dosed to the reactor using two different saturators with Argon (Ar) as carrier gas. After the reaction, the samples were evacuated and transferred to the analysis chamber. All XPS spectra were referenced to Zr3d, settled at 182.2 eV, and spectra treatment was carried out using the CasaXPS software (Version 2.3.16 PR 1.6, Casa Software Ltd, Teignmouth, Devon, UK)

### 4.3. APXPS Experiments

The APXPS experiments were carried out at the near ambient pressure photoemission (NAPP) end station of the CIRCE helical undulator beamline (100–2000 eV photon energy range) at the ALBA Synchrotron Light Facility (Cerdanyola del Vallès, Spain) The spectra were measured with a PHOIBOS 150 NAP analyzer (SPECS GmbH, Berlin, Germany) from SPECS. This analyzer is equipped with four differentially pumped stages connected through small apertures. The electrons photoemitted are focused through the apertures by electrostatic lenses in order to maximize the transmission. This set-up allows the sample is at a maximum pressure of 20 mbar while the detector works in UHV conditions (approx. $1 \times 10^8$ mbar).

For the XPS core level measurements, the beamline optics exit slit was been set to 20 μm and the analyzer pass energy to 20 eV. The total energy resolution of beamline plus analyzer in the measurement conditions was better than 0.3 eV. Incident photon energies of 500 eV for Pd3d and Zr3p and 1150 eV for Cu2p were used, resulting in a sample probing depth of 1.6 nm. Spectra analysis were performed using CasaXPS software utilizing a Shirley background. Peaks fits were achieved using asymmetric functional Lorentzian line shapes for metallic species and symmetric Gaussian–Lorentzian line shapes for non-metallic species. More details about the fitting and the relative quantification procedure are detailed in the Supplementary Materials.

The catalysts (100 mg) were pelletized and mounted onto the sample holder using a resistive button heater for catalysts. The temperature was measured with a K-type thermocouple in direct contact with the catalyst. At first, the samples were reduced in $H_2$ atmosphere during 2 h at 300 °C. $H_2$ gas (1 mbar) was dosed into the analysis chamber by UHV leak valve. After the heating process at 300 °C, the catalysts were cooled down to 100 °C (in $H_2$). At 100 °C, the $H_2$ dosing was stopped and the analysis chamber was evacuated. At this temperature, 1.0 mbar MeOH (methanol, Reg. Ph. Eur, Panreac, purity ≥ 99.8%) and 1.5 mbar $H_2O$ (LC-MS Ultra CHROMASOLV, Fluka, Sigma-Aldrich) were fed into the analysis chamber. The sample was heated until 180 °C (reaction temperature) for 4 h. The data were analyzed using the Casa XPS software. Reaction products were analyzed using a Microvision-IP quadrupole residual gas analyzer from MKS instruments (MKS instruments, Andover, MA, USA), located at the second pumping stage of the XPS analyzer. The m/z values used in the identification of each product were 31 for MeOH, 2 for $H_2$, 18 for $H_2O$, 44 for $CO_2$, and 28 for CO. A MS blank experiment was done with an empty sample holder in order to discard possible catalytic effect of the sample holder filament in the reaction conditions used.

### 4.4. IR Studies

IR spectra were recorded with a Nexus 8700 FTIR spectrometer (Thermo Fisher Scientific, Waltham, MA, USA) using a deuterated-triglycine sulfate (DTGS) detector and acquiring at 4 $cm^{-1}$ resolution. For IR studies a home-made IR catalytic cell which allows in situ treatments at controlled temperatures (from −176 to 500 °C) and atmospheres (vacuum to 1 bar) has been used. The catalysts were pressed into self-supported wafers of 10 mg/$cm^2$. For the reduction experiments, $H_2$ (10 mL·$min^{-1}$) was flowed at 300 °C for 3 h. After the reduction, the catalysts were submitted to a vacuum treatment ($1\cdot10^{-4}$ mbar) at the same temperature (300 °C) for 1 h. Afterwards, the samples were cooled down to −160 °C under dynamic vacuum and CO was then dosed at increasing pressures (0.2–8.0 mbar). Spectra analysis was done using the Origin software. For MSR studies, the samples were initially reduced at 300 °C in $H_2$ flow (10 mL·$min^{-1}$) for 3 and cooling down to 180 °C in $H_2$ flow. At this temperature, MeOH and

water (MeOH:H$_2$O 1:1.5) were fed into the IR cell for 1 h using two different saturators using Argon (17 mL·min$^{-1}$) as carrier gas. Afterwards, the samples were cooled down to $-160$ °C under dynamic vacuum and CO was then dosed at increasing pressures (0.2–8.0 mbar). IR spectra were recorded after each dosage. Reaction products were analyzed using a Balzer mass spectrometer (MS) operating in the multi ion detection mode (MID). The m/z values used in the identification of each product were 31 for MeOH, 2 for H$_2$, 18 for H$_2$O, 44 for CO$_2$, and 28 for CO.

## 5. Conclusions

In this work, the combination of several spectroscopic tools, and specifically the use of surface sensitive techniques, like XPS and in situ IR, has been shown to be important in order to achieve a better picture of surface sites and of their electronic properties, which are key in directing the selectivity in catalytic processes, like MSR. Thus, the electronic properties of the metal sites in monometallic Pd/ZrO$_2$ and bimetallic PdCu/ZrO$_2$ catalysts have been studied using CO as probe molecule in IR studies, and the surface composition of the outermost layers has been studied by APXPS. Using that technique combination, it has been shown that the structure of the ZrO$_2$ support (monoclinic and cubic) influences not only the metal dispersion but also the mobility and reorganization of metal sites under reaction conditions, in both monometallic and bimetallic samples, with a strong influence on the catalytic activity. In addition, spectroscopy with in-situ MS analysis reveals that the effective intermixing between Cu and Pd metals is imperative in optimizing the selectivity in MSR reaction toward CO-free H$_2$ production.

**Supplementary Materials:** The following are available online at http://www.mdpi.com/2073-4344/10/9/1005/s1, Figure S1: Pd3d + Zr3p core lines acquired with hν = 1486 eV of: (**a**) post-reduction and post-reaction (MSR) Pd/monoclinic ZrO$_2$ catalyst (**b**) post-reduction and post-reaction (MSR) Pd/cubic ZrO$_2$ catalyst; Figure S2a: XRD of reduced Pd/ZrO$_2$-m (red) and Pd/ZrO$_2$-c (blue) catalysts; Figure S2b: XRD of reduced PdCu/ZrO$_2$-m (red) and PdCu/ZrO$_2$-c (blue) catalysts; Figure S3: MS spectra of (**a**) PdCu/ZrO$_2$-m and (**b**) PdCu/ZrO$_2$-c under MSR in the NAP-XPS study; Figure S4: (**a**) O1s gas phase spectra acquired with hν= 700 eV and (**b**) C1s gas phase spectra acquired with hν = 500 eV under MSR conditions (2h). PdCu/ZrO$_2$-m (green) y PdCu/ZrO$_2$-c (purple); Figure S5: Zr3d core lines acquired with hν = 500 eV of reduced PdCu/ZrO$_2$-m catalyst (red) and under reaction conditions (MSR); Figure S6: Pd3d + Zr3p core line acquired with hν = 773 eV under reaction conditions (MSR) of PdCu/ZrO$_2$-m catalyst; Figure S7: IR spectra of CO adsorption at increasing CO dosing on reduced (**a**) ZrO$_2$-m and (**b**) ZrO$_2$-c; Figure S8 IR spectra deconvolution of (**a**) Pd/ZrO$_2$-m reduced; (**b**) Pd/ZrO$_2$-m MSR; (**c**) Pd/ZrO$_2$-c reduced (**d**) Pd/ZrO$_2$-c MSR; Figure S9: IR spectra of CO adsorption on reduced catalysts (red) and after MSR (blue). (**a**) Pd/ZrO$_2$-m and (**b**) Pd/ZrO$_2$-c; Figure S10: IR spectra (1700–1300 cm$^{-1}$) of the Pd/ZrO$_2$-m (red) and Pd/ZrO$_2$-c (blue) after MSR; Figure S11: MS spectra of (**a**) PdCu/ZrO$_2$-m and (**b**) PdCu/ZrO$_2$-c under MSR in the in-situ IR studies; Table S1: Pd3d$_{5/2}$ and Zr3p$_{3/2}$ binding energy (BE, eV) acquired with hν = 1486 eV and surface chemical composition (atomic ratio) of Pd/monoclinic ZrO$_2$ catalyst and Pd/cubic ZrO$_2$ catalyst; Table S2: Physico-chemical characterization of the monometallic and bimetallic samples; Table S3: Values derived from the mass spectra of the catalysts under MSR in the NAP-XPS study (partial pressures ratio); Table S4: Values derived from the mass spectra of the catalysts under MSR in the in-situ IR study (partial pressures ratio).

**Author Contributions:** Designed and performed the experiments, data curation and writing-original draft preparation, P.C.; Performed experiments, data curation, and writing—review and editing, V.P.-D.; Data curation, performed experiments, and writing—editing, D.R.; Performed experiments, B.M.P.; Review and samples, A.M., C.A. and C.M.-P. All authors have read and agreed to the published version of the manuscript.

**Funding:** The research leading to these results has received funding from European Research council project SYNCATMATCH (671093) and from Spanish Ministry of Science, Innovation and Universities with the project "I + D + I research challenges (RTI2018-099668-B-C21)". This work also was financially supported by: Base Funding-UIDB/00511/2020 of the Laboratory for Process Engineering, Environment, Biotechnology and Energy—LEPABE-funded by national funds through the FCT/MCTES (PIDDAC); European Union's Seventh Framework Program (FP/2007-2013) for the Fuel Cells and Hydrogen Joint Technology Initiative under grant agreement no. 303476.

**Acknowledgments:** The authors are thankful for the support of the ALBA Synchrotron Light Source staff for the successful performance of the measurements at CIRCE beamline (BL 24). This work has been done in the framework of the doctorate in Materials Science of the Universitat Autònoma de Barcelona.

**Conflicts of Interest:** The authors declare no conflict of interest.

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
