# Peer review of "Influence of the ZrO2 Crystalline Phases on the Nature of Active Sites in PdCu/ZrO2 Catalysts for the Methanol Steam Reforming Reaction—An In Situ Spectroscopic Study"

_catalysts, doi:10.3390/catal10091005_

Round 1
Reviewer 1 Report
This paper presents a study in which regular vacuum XPS and ambient pressure XPS (APXPS) are used to characterize the chemical state and structure of PdCu particles supported on both monoclinic and cubic ZrO2 under methanol steam reforming reaction conditions. FTIR of adsorbed CO was also used to help characterize the types of sites on the surface of the catalysts. They do a very thorough job analyzing the XPS and IR data and ultimately conclude that “the structure of the ZrO2 support (monoclinic and cubic) influences not only the metal dispersion but also the mobility and reorganization of metal sites under reaction conditions, in both monometallic and bimetallic samples, with a strong influence on the catalytic activity.” While their data and analysis provides reasonable support for these conclusions, in my opinion, the paper suffers from a lack of analysis of the thermodynamics of the Pd-Cu system and what would be expected for the structure of supported PdCu particles. Specific issues related to this that the authors should address in the final version of the paper are listed below.
- How big are the metal particles? The paper mentions that there are particle size differences between the two supports but never gives an estimate of the actual particle sizes. It would also be useful to report the surface area of the zirconia supports that were used. The use of high metal weight loadings (4 wt.% for Pd/ZrO2 and 4 wt.% Pd, 20 wt.% for PdCu/ZrO2) coupled with a support which likely had relatively low surface area, means the supported metal particles must have been quite large. If this is the case I could believe that the support provides some modest influence over the morphology of the metal particles, but any electronic effects would be minimal and the thermodynamics (energetics) of the metal particles themselves should be largely independent of the support. This argues against some of their support effects conclusions.
- What do bulk thermodynamics predict for Pd-Cu mixtures? Phase diagrams for Pd-Cu mixtures are readily available in the literature and I suspect that they show that Pd and Cu are totally miscible for the conditions used in this study (I could be wrong about this). If this is indeed the case and the metal particles are large, it would be surprising that the support would influence phase separation in the particles which they seem to be arguing has occurred at least under some conditions. Their conclusions are still reasonable, but these factors also need to be addressed in the analysis.
- Enrichment of one of the metals on the surface of the metal particles under reaction conditions is certainly possible and for the mildly oxidizing conditions for MSR one would expect the more easily oxidized metal to be segregate to the surface. Is that what is observed here? Perhaps this is what they are implying when they refer to the electronegativities of the metals. A little more explanation would be useful here.
Reviewer 2 Report
Present article is very good written and can be interesting for the scientists working in different spheres of chemistry and material science. Experiments are rather good described and their results are well discussed.
Nevertheless some questions have been arised:
- Why authors do not provide the whole spectrum of the catalysts to see the lines of Y, which was used to stabilize the structure?
- Why authors do not provide any TEM images and XRD data to confirm the structure formation ?
